# Optimizing Data Usage via Differentiable Rewards

## Abstract

To acquire a new skill, humans learn better and faster if a tutor, based on their current knowledge level, informs them of how much attention they should pay to particular content or practice problems. Similarly, a machine learning model could potentially be trained better with a scorer that "adapts" to its current learning state and estimates the importance of each training data instance. Training such an adaptive scorer efficiently is a challenging problem; in order to precisely quantify the effect of a data instance at a given time during the training, it is typically necessary to first complete the entire training process. To efficiently optimize data usage, we propose a reinforcement learning approach called Differentiable Data Selection (DDS). In DDS, we formulate a scorer network as a learnable function of the training data, which can be efficiently updated along with the main model being trained. Specifically, DDS updates the scorer with an intuitive reward signal: it should up-weigh the data that has a similar gradient with a dev set upon which we would finally like to perform well. Without significant computing overhead, DDS delivers strong and consistent improvements over several strong baselines on two very different tasks of machine translation and image classification.[1]

## 1 Introduction

While deep learning models are remarkably good at fitting large data sets, their performance is also highly sensitive to the structure and domain of their training data. Training on out-of-domain data can lead to worse model performance, while using more relevant data can assist transfer learning. Previous work has attempted to create strategies to handle this sensitivity by selecting subsets of the data to train the model on (Jiang & Zhai, 2007; Wang et al.; Axelrod et al., 2011; Moore & Lewis, 2010), providing different weights for each example (Sivasankaran et al., 2017; Ren et al., 2018), or changing the presentation order of data (Bengio et al., 2009; Kumar et al., 2019).

However, there are several challenges with the existing work on better data usage strategies. Most work data filtering criterion or training curriculum rely on domain-specific knowledge and hand-designed heuristics, which can be sub-optimal. To avoid hand designed heuristics, several works propose to optimize a parameterized neural network to learn the data usage schedule, but most of them are tailored to specific use cases, such as handling noisy data for classification (Jiang et al., 2018), learning a curriculum learning strategy for NMT (Kumar et al., 2019), and actively selecting data for annotation (Fang et al., 2017; Wu et al., 2018). Fan et al. (2018) proposes a more general teacher-student framework that first trains a teacher network to select data that directly optimizes development set accuracy over multiple training runs. However, because running multiple runs of training simply to train this teacher network entails an $n$-fold increase in training time for $n$ runs, this is infeasible in many practical settings. In addition, in preliminary experiments we also found the single reward signal provided by dev set accuracy at the end of training noisy to the extent that we were not able to achieve results competitive with simpler heuristic training methods.

In this paper, we propose an alternative: a general Reinforcement Learning (RL) framework for optimizing training data usage by training a *scorer network* that minimizes the model loss on the development set. We formulate the scorer network as a function of the current training examples only, making it possible to re-use the model architecture which is designed and trained for the main task.

---

[1]We will make the code publicly available upon acceptance.

Thus, our method requires no heuristics and is generalizable to various tasks. To make the scorer adaptive, we perform frequent and efficient updates of the scorer network using a reward function inspired by recent work on learning using data from auxiliary tasks (Du et al., 2018; Liu et al., 2019b), which use the similarity between two gradients as a measure of task relevance. We propose to use the gradient alignment between the training examples and the dev set as a reward signal for *a parametric scorer network*, as illustrated in Figure 1. We then formulate our framework as an optimization problem found in many prior works such as meta-learning (Finn et al., 2017), noisy data filtering (Ren et al., 2018), and neural architecture search (Liu et al., 2019a), and demonstrate that our proposed update rules follow a direct differentiation of the scorer parameters to optimize the model loss on the dev set. Thus we refer to our framework as "Differentiable Data Selection" (DDS).

We demonstrate two concrete instantiations of the DDS framework, one for a more general case of image classification, and the other for a more specific case of neural machine translation (NMT). For image classification, we test on both CIFAR-10 and ImageNet. For NMT, we focus on a multilingual setting, where we optimize data usage from a multilingual corpus to improve the performance on a particular language. For these two very different and realistic tasks, we find the DDS framework brings significant improvements over the baselines for all settings.

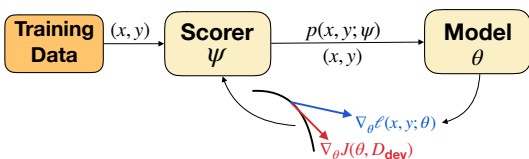

**Figure 1:** The general workflow of DDS.

## 2  DIFFERENTIABLE DATA SELECTION

### 2.1  RISK, TRAINING, AND DEVELOPMENT SETS

Commonly in machine learning, we seek to find the parameters $\theta^*$ that minimize the *risk* $J(\theta, P)$, the expected value of a loss function $\ell(x, y; \theta)$, where $\langle x, y \rangle$ are pairs of inputs and associated labels sampled from a particular distribution $P(X, Y)$:

$$\theta^* = \operatorname*{argmin}_{\theta} J(\theta, P) \quad \text{where} \quad J(\theta, P) = \mathbb{E}_{x, y \sim P(X, Y)}[\ell(x, y; \theta)] \tag{1}$$

Ideally, we would like the risk $J(\cdot)$ to be minimized over the data distribution that our system sees at test time, ie. $P_{\text{test}}(X, Y)$. Unfortunately, this distribution is unknown at training time, so instead we collect a training set $\mathcal{D}_{\text{train}} = \{(x_i, y_i) : i = 1, ..., N_{\text{train}}\}$ with distribution $P_{\text{train}}(X, Y) = \text{Uniform}(\mathcal{D}_{\text{train}})$, and minimize the *empirical risk* by taking $\langle x, y \rangle \sim P_{\text{train}}(X, Y)$. Since we need a sufficiently large training set $\mathcal{D}_{\text{train}}$ to train a good model, it is hard to ensure that $P_{\text{train}}(X, Y) \approx P_{\text{test}}(X, Y)$. In fact, we often accept that training data comes from a different distribution than test data. The discrepancy between $P_{\text{train}}(X, Y)$ and $P_{\text{test}}(X, Y)$ manifests itself in the form of problems such as overfitting (Zhang et al., 2017; Srivastava et al., 2014), covariate shift (Shimodaira, 2000), and label shift (Lipton et al., 2018).

However, unlike the large training set, we can collect a relatively small development set $\mathcal{D}_{\text{dev}} = \{(x_i, y_i) : i = 1, ..., N_{\text{dev}}\}$ with distribution $P_{\text{dev}}(X, Y)$ which is much closer to $P_{\text{test}}(X, Y)$[2]. Since $\mathcal{D}_{\text{dev}}$ is a better approximation of our test-time scenario[3], we can use $\mathcal{D}_{\text{dev}}$ to get reliable feedback to learn to better utilize our training data from $\mathcal{D}_{\text{train}}$. In particular, we propose to train a *scorer* network, parameterized by $\psi$, to provide guidance on training data usage to minimize $J(\theta, \mathcal{D}_{\text{dev}})$ .

### 2.2  REINFORCEMENT LEARNING FOR OPTIMIZING DATA USAGE

We propose to optimize the scorer's parameters $\psi$ in an RL setting. Our *environment* is the model state $\theta$ and an example $\langle x, y \rangle$. Our RL *agent* is the scorer network $\psi$, which optimizes the data usage

---

[2]As is standard in machine learning experiments, we make sure that $\mathcal{D}_{\text{dev}}$ has no overlap with $\mathcal{D}_{\text{train}}$ or $\mathcal{D}_{\text{test}}$. Details of how we construct the $\mathcal{D}_{\text{dev}}$ can be found in Appendix A.2 and A.4.

[3]For example, in Section 3.2 we would like to use training data from many different languages to improve the performance of a particular low-resource language. Here we can gather a small set of $\mathcal{D}_{\text{dev}}$ from the low resource language, even if we can't gather a large training set in the language. Moreover, in a domain adaptation setting we can obtain a small dev set in the target domain, or in a setting of training on noisy data we can often obtain a small clean dev set.

for the current model state. The agent's *reward* on picking an example approximates the dev set performance of the resulting model after the model is updated on this example.

Our scorer network is parameterized as a differentiable function that only takes as inputs the features of the example $\langle x, y \rangle$. Intuitively, it represents a distribution over the training data where more important data has a higher probability of being used, denoted $P(X, Y; \psi)$. Unlike prior methods which generally require complicated featurization of both the model state and the data as input to the RL agent (Fan et al., 2018; Jiang et al., 2018; Fang et al., 2017), our formulation is much simpler and generalizable to different tasks. Since our scorer network does not consider the model parameters $\theta_t$ as input, we update it iteratively with the model so that at training step $t$, $P(X, Y; \psi_t)$ provides an up-to-date data scoring feedback for a given $\theta_t$.

Although the above formulation is simpler and more general, it requires much more frequent updates to the scorer parameter $\psi$. Existing RL frameworks simply use the change in dev set risk as the regular reward signal, which makes the update expensive and unstable (Fan et al., 2018; Kumar et al., 2019). Therefore, we propose a novel reward function as an approximation to $\Delta J_{\text{dev}}(x, y)$ to quantify the effect of the training example $\langle x, y \rangle$. Inspired by Du et al. (2018) (which uses gradient similarity between two tasks to measure the adaptation effect between them, we use the agreement between the model gradient on data $\langle x, y \rangle$ and the gradient on the dev set to approximate the effect of $\langle x, y \rangle$ on dev set performance. This reward simply implies that we prefer data that moves $\theta$ in the direction that minimizes the dev set risk:

$$R(x, y) = \Delta J_{\text{dev}}(x, y) \approx \nabla_\theta \ell(x, y; \theta_{t-1})^\top \cdot \nabla_\theta J(\theta_t, \mathcal{D}_{\text{dev}}) \tag{2}$$

According to the REINFORCE algorithm (Williams, 1992), the update rule for $\psi$ is thus

$$\psi_{t+1} \leftarrow \psi_t + \underbrace{\nabla_\theta \ell(x, y; \theta_{t-1}) \cdot \nabla_\theta J(\theta_t, \mathcal{D}_{\text{dev}})}_{R(x,y)} \nabla_\psi \log(P(X, Y; \psi)) \tag{3}$$

The update rule for the model is simply

$$\theta_t \leftarrow \theta_{t-1} - \nabla_\theta J(\theta_{t-1}, P(X, Y; \psi)) \tag{4}$$

For simplicity of notation, we omit the learning rate term. Full derivation can be found in Appendix A.1. By alternating between Eqn. 4 and Eqn. 3, we can iteratively update $\theta$ using the guidance from the scorer network, and update $\psi$ to optimize the scorer using feedback from the model.

Our formulation of scorer network as $P(X, Y; \psi)$ has several advantages. First, it provides the flexibility that we can either sample a training instance or equivalently scale the update from the training instance based on its score. Specifically, we provide an algorithm under the DDS framework for multilingual NMT (see Sec. 3.2), where the former is more efficient, and another more general algorithm for image classification (see Sec. 3.1), where the latter choice is natural. Second, it allows easy integration of prior knowledge of the data, which is shown to be effective in Sec. 4.

## 2.3 DERIVING REWARDS THROUGH DIRECT DIFFERENTIATION

In this section, we show that the update for the scorer network in Eqn. 3 can be approximately derived as the solution of a bi-level optimization problem (Colson et al., 2007), which has been applied to many different lines of research (Baydin et al., 2018; Liu et al., 2019a; Ren et al., 2018).

Under our framework, the scorer samples the data by $\langle x, y \rangle \sim P(X, Y; \psi)$, and $\psi$ will be chosen so that $\theta^*$ that optimizes $J(\theta, P(X, Y; \psi))$ will approximately minimize $J(\theta, P_{\text{dev}}(X, Y))$:

$$\psi^* = \operatorname*{argmin}_\psi J(\theta^*(\psi), \mathcal{D}_{\text{dev}}) \text{ where } \theta^*(\psi) = \operatorname*{argmin}_\theta \mathbb{E}_{x,y \sim P(X,Y;\psi)} [\ell(x, y; \theta)] \tag{5}$$

The connection between $\psi$ and $\theta$ in Eqn. 5 shows that $J(\theta_t, \mathcal{D}_{\text{dev}})$ is differentiable with respect to $\psi$. Now we can approximately compute the gradient $\nabla_\psi J(\theta_t, \mathcal{D}_{\text{dev}})$ as follows:

$$
\begin{aligned}
\nabla_\psi J(\theta_t, \mathcal{D}_{\text{dev}}) &= \nabla_{\theta_t} J(\theta_t, \mathcal{D}_{\text{dev}})^\top \cdot \nabla_\psi \theta_t(\psi) && \text{(chain rule)} \\
&= \nabla_{\theta_t} J(\theta_t, \mathcal{D}_{\text{dev}})^\top \cdot \nabla_\psi (\theta_{t-1} - \nabla_\theta J(\theta_{t-1}, \psi)) && \text{(substitute } \theta_t \text{ from Eqn 4)} \\
&\approx -\nabla_{\theta_t} J(\theta_t, \mathcal{D}_{\text{dev}})^\top \cdot \nabla_\psi (\nabla_\theta J(\theta_{t-1}, \psi)) && \text{(assume } \nabla_\psi \theta_{t-1} \approx 0) \\
&= -\nabla_\psi \mathbb{E}_{x,y \sim P(X,Y;\psi)} \left[ \nabla_\theta J(\theta_t, \mathcal{D}_{\text{dev}})^\top \cdot \nabla_\theta \ell(x, y; \theta_{t-1}) \right] \\
&= -\mathbb{E}_{x,y \sim P(X,Y;\psi)} \left[ \left( \nabla_\theta J(\theta_t, \mathcal{D}_{\text{dev}})^\top \cdot \nabla_\theta \ell(x, y; \theta_{t-1}) \right) \cdot \nabla_\psi \log P(x, y; \psi) \right]
\end{aligned}
\tag{6}
$$

Here, we make a Markov assumption that $\nabla_\psi \theta_{t-1} \approx 0$, assuming that at step $t$, given $\theta_{t-1}$ we do not care about how the values of $\psi$ from previous steps led to $\theta_{t-1}$. Eqn. 9 leads to a rule to update $\psi$ using gradient descent, which is exactly the same as the RL update rule in Eqn. 3.

Note that our derivation above does not take into the account that we might use different optimizing algorithms, such as SGD or Adam (Kingma & Ba, 2015), to update $\theta$. We provide detailed derivations for several popular optimization algorithms in Appendix A.1.

One potential concern with our approach is that because we optimize $\psi_t$ directly on the dev set using $J(\theta_t, \mathcal{D}_{\text{dev}})$, we may risk indirectly overfitting model parameters $\theta_t$ by selecting a small subset of data that is overly specialized. However we do not observe this problem in practice, and posit that this because (1) the influence of $\psi_t$ on the final model parameters $\theta_t$ is quite indirect, and acts as a "bottleneck" which has similarly proven useful for preventing overfitting in neural models Grézl et al. (2007), and (2) because the actual implementations of DDS (which we further discuss in Section 3) only samples a subset of data from $\mathcal{D}_{\text{train}}$ at each optimization step, further limiting expressivity.

## 3 CONCRETE INSTANTIATIONS OF DDS

We now turn to discuss two concrete instantiations of DDS that we use in our experiments: a more generic example of classification, which should be applicable to a wide variety of tasks, and a specialized application to the task of multilingual NMT, which should serve as an example of how DDS can be adapted to the needs of specific applications.

### 3.1 FORMULATION FOR CLASSIFICATION

---

**Algorithm 1:** Training a classification model with DDS.

**Input** : $\mathcal{D}_{\text{train}}, \mathcal{D}_{\text{dev}}$
**Output :** Optimal parameters $\theta^*$

1 Initializer $\theta_0$ and $\psi_0$
2 **for** $t = 1$ **to** *num_train_steps* **do**
3      Sample $B$ training data points $x_i, y_i \sim \text{Uniform}(\mathcal{D}_{\text{train}})$
4      Sample $B$ validation data points $x_i', y_i' \sim \text{Uniform}(\mathcal{D}_{\text{dev}})$
     ▷ *Optimize $\theta$*
5      Update $\theta_t \leftarrow \text{GradientUpdate}\left(\theta_{t-1}, \sum_{i=1}^{B} p(x_i, y_i; \psi_{t-1}) \nabla_\theta \ell(x_i, y_i; \theta_{t-1})\right)$
     ▷ *Evaluate $\theta_t$ on $\mathcal{D}_{dev}$*
6      Let $d_\theta \leftarrow \frac{1}{B} \sum_{j=1}^{B} \nabla_\theta \ell(x_j', y_j'; \theta_t)$
     ▷ *Optimize $\psi$*
7      Let $d_\psi \leftarrow \frac{1}{B} \sum_{i=1}^{B} \left[\left(d_\theta^\top \cdot \nabla_\theta \ell(x_i, y_i; \theta_{t-1})\right) \cdot \nabla_\psi \log p(x_i, y_i; \psi)\right]$
8      Update $\psi_t \leftarrow \text{GradientUpdate}(\psi_{t-1}, d_\psi)$
9 **end**

---

Algorithm 1 presents the pseudo code for the training process on classification tasks, using the notations introduced in Section 2. The main classification model is parameterized by $\theta$. The scorer $p(X, Y; \psi)$ is an identical network with the main model, but with independent weights, *i.e.* $p(X, Y; \psi)$ does not share weights with $\theta$. For each example $x_i$ in a minibatch uniformly sampled from $\mathcal{D}_{\text{train}}$, this DDS model outputs a scalar from the data $x_i$. All scalars are passed through a softmax function to compute the relative probabilities of the examples in the minibatch, and their gradients are scaled accordingly when applied to $\theta$. Note that our actual formulation of $p(X, Y; \psi)$ does *not* depend on $Y$, but we keep $Y$ in the notation for consistency with the formulation of the DDS framework. Note that we have two gradient update steps, one for the model parameter $\theta_t$ in Line 5 and the other for the DDS scorer parameter $\psi$ in Line 8. For the model parameter update, we can simply use any of the standard optimization update rule. For the scorer $\psi$, we use the update rule derived in Section 2.3.

**Per-Example Gradient.** As seen from Line 7 of Algorithm 1, as well as from Eqn. 13, DDS requires us to compute $\nabla_\theta \ell(x_i, y_i; \theta_{t-1})$, *i.e.* the gradient for each example in a batch of training data. This operation is very slow and memory intensive, especially when the batch size is large, *e.g.* our experiments on ImageNet use a batch size of 4096 (see Section 4). Therefore, we propose an efficient approximation of this per-example gradient computation via the first-order Taylor expansion of $\ell(x_i, y_i; \theta_{t-1})$. In particular, for any vector $v \in \mathbb{R}^{|\theta|}$, with sufficiently small $\epsilon > 0$, we have:

$$v^\top \cdot \nabla_\theta \ell(x_i, y_i; \theta_{t-1}) \approx \frac{1}{\epsilon} \left(\ell(x_i, y_i; \theta_{t-1} + \epsilon v) - \ell(x_i, y_i; \theta_{t-1})\right), \tag{7}$$

Eqn 7 can be implemented by keeping a shadow version of parameters $\theta_{t-1}$, caching training loss $\ell(x_i, y_i; \theta_{t-1})$, and computing the new loss with $\theta_{t-1} + \epsilon v$. Here, $v$ is $d_\theta$ as in Line 7 of Algorithm 1.

## 3.2 FORMULATION FOR MULTILINGUAL NMT

Next we demonstrate an application of DDS to multilingual models for NMT, specifically for improving accuracy on low-resource languages (LRL) (Zoph et al., 2016; Neubig & Hu, 2018). In this setting, we assume that we have a particular LRL $S$ that we would like to translate into target language $T$, and we additionally have a multilingual corpus $\mathcal{D}_{\text{train}}$ that has parallel data between $n$ source languages $(S_1, S_2, ..., S_n)$ and target language $T$. We would like to pick parallel data from any of the source languages to the target language to improve translation of a particular LRL $S$, so we assume that $\mathcal{D}_{\text{dev}}$ exclusively consists of parallel data between $S$ and $T$. Thus, DDS will attempt to select data from $\mathcal{D}_{\text{train}}$ that improve accuracy on $S$-to-$T$ translation as represented by $\mathcal{D}_{\text{dev}}$.

---

**Algorithm 2:** Training multilingual NMT with DDS.

**Input** : $\mathcal{D}_{\text{train}}$; K: number of data to train the NMT model before updating $\psi$; E: number of updates
for $\psi$; $\alpha_1, \alpha_2$: discount factors for the gradient
**Output :** The converged NMT model $\theta^*$

1   Initialize $\psi_0, \theta_0$
  ▷ *Initialize the gradient of each source language*
2   $grad[S_i] \leftarrow 0$ **for** *i in n*
3   **while** $\theta$ *not converged* **do**
4      $X, Y \leftarrow$ load_data$(\psi, \mathcal{D}_{\text{train}}, K)$
     ▷ *Train the NMT model*
5      **for** $x_i, y$ *in* $X, Y$ **do**
6         $\theta_t \leftarrow$ GradientUpdate $\left(\theta_{t-1}, \nabla_{\theta_{t-1}} \ell(x_i, y; \theta_{t-1})\right)$
7         $grad[S_i] \leftarrow \alpha_1 \times grad[S_i] + \alpha_2 \times \nabla_{\theta_{t-1}} \ell(x_i, y; \theta_{t-1})$
8      **end**
     ▷ *Optimize $\psi$*
9      **for** *iter in E* **do**
10         sample $B$ data pairs from $\mathcal{D}_{\text{train}}$
11         $d_\psi \leftarrow \frac{1}{B} \sum_{j=1}^{B} \sum_{i=1}^{n} \left[ grad[S_i]^\top grad[S] \cdot \nabla_{\psi_{t-1}} \log\left(p\left(S_i | y_j; \psi_{t-1}\right)\right) \right]$
12         $\psi_t \leftarrow$ GradientUpdate$(\psi_{t-1}, d_{\psi_{t-1}})$
13      **end**
14   **end**

---

To make training more efficient and stable in this setting, we make three simple modifications of the main framework in Section 2.3 that take advantage of the problem structure of multilingual NMT. First, instead of directly modeling $p(X, Y; \psi)$, we assume a uniform distribution over the target sentence $Y$, and only parameterize the conditional distribution of which source language sentence to pick given the target sentence: $p(X | y; \psi)$. This design follows the formulation of Target Conditioned Sampling (TCS; Wang & Neubig (2019)), an existing state-of-the-art data selection method that uses a similar setting but models the distribution $p(X | y)$ using heuristics. Since the scorer only needs to model a simple distribution over training languages, we use a fully connected 2-layer perceptron network. Second, we only update $\psi$ after updating the NMT model for a fixed number of steps. Third, we sample the data according to $p(X | y; \psi)$ to get a Monte Carlo estimate of the objective in Eqn. 5. This significantly reduces the training time compared to using all data. The pseudo code of the training process is in Algorithm 2.

## 4 EXPERIMENTS

We now discuss experimental results on both image classification, an instance of the general classification problem using Algorithm 1, and multilingual NMT using Algorithm 2.

### 4.1 EXPERIMENTAL SETTINGS

**Data.** We apply our method on established benchmarks for image classification and multilingual NMT. For image classification, we use CIFAR-10 (Krizhevsky, 2009) and ImageNet (Russakovsky et al., 2015). For each dataset, we consider two settings: a reduced setting where only roughly 10% of the training labels are used, and a full setting, where all labels are used. Specifically, the reduced setting for CIFAR-10 uses the first 4000 examples in the training set, and with ImageNet, the reduced setting uses the first 102 TFRecord shards as pre-processed by Kornblith et al. (2019). We use the size of $224 \times 224$ for ImageNet.

**Table 1:** Results for image classification accuracy (left) and multilingual MT BLEU (right). For MT, the statistical significance is indicated with $*$ (p < 0.005) and † (p < 0.0001).

| Methods | CIFAR-10 (WRN-28-$k$) | | ImageNet (ResNet-50) | |
|---|---|---|---|---|
| | 4K, $k = 2$ | Full, $k = 10$ | 10% | Full |
| Uniform | 82.60±0.17 | 95.55±0.15 | 56.36/79.45 | 76.51/93.20 |
| SPCL | 81.09±0.22 | 93.66±0.12 | - | - |
| BatchWeight | 79.61±0.50 | 94.11±0.18 | - | - |
| MentorNet | 83.11±0.62 | 94.92±0.34 | - | - |
| DDS | 83.63± 0.29 | 96.31± 0.13 | **56.81/79.51** | **77.23/93.57** |
| retrained DDS | **85.56±0.20** | **97.91±0.12** | - | - |

| Methods | aze | bel | glg | slk |
|---|---|---|---|---|
| Uniform | 10.31 | 17.21 | 26.05 | 27.44 |
| SPCL | 9.07 | 16.99 | 23.64 | 21.44 |
| Related | 10.34 | 15.31 | 27.41 | 25.92 |
| TCS | 11.18 | 16.97 | 27.28 | 27.72 |
| DDS | 10.74 | 17.24 | 27.32 | **28.20**$*$ |
| TCS+DDS | **11.84**$*$ | **17.74**† | **27.78** | 27.74 |

For multilingual NMT, we use the 58-language-to-English TED dataset (Qi et al., 2018). Following prior work (Qi et al., 2018; Neubig & Hu, 2018; Wang et al., 2019b), we evaluate translation from four low-resource languages (LRL) Azerbaijani (`aze`), Belarusian (`bel`), Galician (`glg`), and Slovak (`slk`) to English, where each is paired with a similar high-resource language Turkish (`tur`), Russian (`rus`), Portugese (`por`), and Czech (`ces`) (details in Appendix A.3). We combine data from all 8 languages, and use DDS to optimize data selection for each LRL.

**Models and Training Details.** For image classification, on CIFAR-10, we use the pre-activation WideResNet-28 (Zagoruyko & Komodakis, 2016), with width factor $k = 2$ for the reduced setting and $k = 10$ for the normal setting. For ImageNet, we use the post-activation ResNet-50 (He et al., 2016). These implementations reproduce the numbers reported in the literature (Zagoruyko & Komodakis, 2016; He et al., 2016; Xie et al., 2017), and additional details can be found in Appendix A.4.

For NMT, we use a standard LSTM-based attentional baseline (Bahdanau et al., 2015), which is similar to previous models used in low-resource scenarios both on this dataset (Neubig & Hu, 2018; Wang et al., 2019b) and others (Sennrich & Zhang, 2019) due to its relative stability compared to other options such as the Transformer (Vaswani et al., 2017). Accuracy is measured using BLEU score (Papineni et al., 2002). More experiment details are noted in Appendix A.2.

**Baselines and Our Methods.** For both image classification and multi-lingual NMT, we compare the following data selection methods. **Uniform** where data is selected uniformly from all of the data that we have available, as is standard in training models. **SPCL** (Jiang et al., 2015), a curriculum learning method that dynamically updates the curriculum to focus more on the "easy" training examples based on model loss. **DDS**, our proposed method.

For image classification, we compare with several additional methods designed for filtering noisy data on CIFAR-10, where we simply consider the dev set as the clean data. **BatchWeight** (Ren et al., 2018), a method that scales example training loss in a batch with a locally optimized weight vector using a small set of clean data. **MentorNet** (Jiang et al., 2018), a curriculum learning method that trains a mentor network to select clean data based on features from both the data and the main model. For machine translation, we also compare with two state-of-the-art heuristic methods for multi-lingual data selection. **Related** where data is selected uniformly from the target LRL and a linguistically related HRL (Neubig & Hu, 2018). **TCS**, a recently proposed method of "target conditioned sampling", which uniformly chooses target sentences, then picks which source sentence to use based on heuristics such as word overlap (Wang & Neubig, 2019). Note that both of these methods take advantage of structural properties of the multi-lingual NMT problem, and do not generalize to other problems such as classification.

DDS is a flexible framework to incorporate prior knowledge about the data using the scorer network, which can be especially important when the data has certain structural properties such as language or domain. We test such a setting of DDS for both tasks. For image classification, we use **retrained DDS**, where we first train a model and scorer network using the standard DDS till convergence. The trained scorer network can be considered as a good prior over the data, so we use it to train the final model from scratch again using DDS. For multilingual NMT, we experiment with **TCS+DDS**, where we initialize the parameters of DDS with the TCS heuristic, then continue training.

## 4.2 MAIN RESULTS

The results of the baselines and our method are listed in Table 1. First, comparing the standard baseline strategy of "Uniform" and the proposed method of "DDS" we can see that in all 8 settings

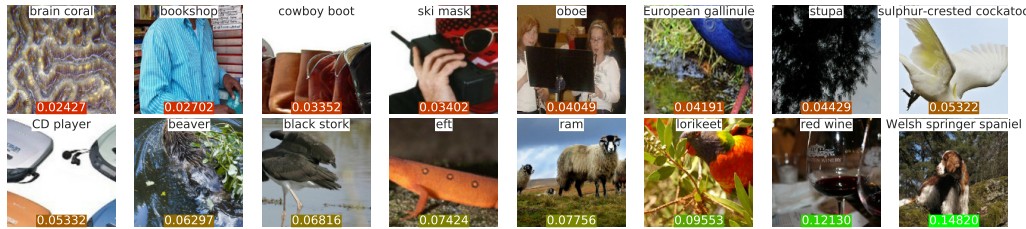

**Figure 3:** Example images from the ImageNet and their weights assigned by DDS.

DDS improves over the uniform baseline. This is a strong indication of both the consistency of the improvements that DDS can provide, and the generality – it works well in two very different settings. Next, we find that DDS outperforms SPCL by a large margin for both of the tasks, especially for multilingual NMT. This is probably because SPCL weighs the data only by their easiness, while ignoring their relevance to the dev set, which is especially important in settings where the data in the training set can have very different properties such as the different languages in multilingual NMT.

DDS also brings improvements over the state-of-the-art intelligent data utilization methods. For image classification, DDS outperforms MentorNet and BatchWeight on CIFAR-10 in all settings. For NMT, in comparison to Related and TCS, vanilla DDS performs favorably with respect to these state-of-the-art data selection baselines, outperforming each in 3 out of the 4 settings (with exceptions of slightly underperforming Related on `glg` and TCS on `aze`). In addition, we see that incorporating prior knowledge into the scorer network leads to further improvements. For image classification, retrained DDS can significantly improve over regular DDS, leading to the new state-of-the-art result on the CIFAR-10 dataset. For mulitlingual NMT, TCS+DDS achieves the best performance in three out of four cases (with the exception of `slk`, where vanilla DDS already outperformed TCS).[4]

DDS does not incur much computational overhead. For image classification and multilingual NMT respectively, the training time is about $1.5\times$ and $2\times$ the regular training time without DDS[5].

### 4.3 ANALYSIS

**Image Classification.** Prior work on heuristic data selection has found that the model performs better if we feed higher quality or more domain-relevant data towards the end of training (van der Wees et al., 2017; Wang et al., 2019a). Here we verify this observation by analyzing the learned importance weight at the end of training for image classification. Figure 2 shows that at the end of training, DDS learns to balance the class distribution, which is originally unbalanced due to the dataset creation. Figure 3 shows that at the

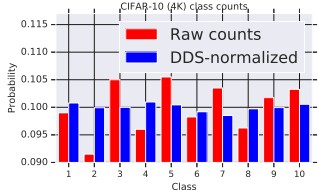

**Figure 2:** Class distributions of CIFAR-10 4K.

end of training, DDS assigns higher probabilities to images with clearer class content from ImageNet. These results show that DDS learns to focus on higher quality data towards the end of training.

**NMT.** Next, we focus on multi-lingual NMT, where the choice of data directly corresponds to picking a language, which has an intuitive interpretation. Since DDS adapts the data weights dynamically to the model throughout training, here we analyze how the dynamics of learned weights.

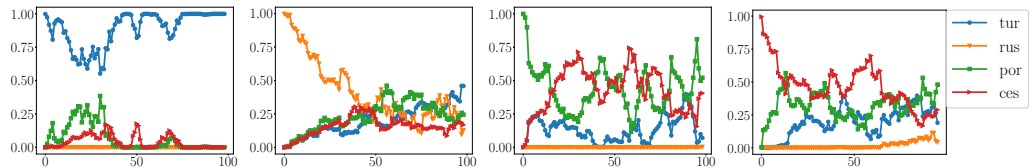

**Figure 4:** Language usage for TCS+DDS by training step. *From left to right*: `aze`, `bel`, `glg`, `slk`.

We plot the probability distribution of the four HRLs (because they have more data and thus larger impact on training) over the course of training. Figure 4 shows the change of language distribution

---

[4]For the NMT significance tests (Clark et al., 2011) find significant gains over the baseline for aze, slk, and bel. For glg the gain is not significant, but DDS-uniform without heuristics performs as well as the TCS baseline.

[5]The code for multilingual NMT is not optimized, so its training time could be reduced further

for TCS+DDS. Since TCS selects the language with the largest vocabulary overlap with the LRL, the distribution is initialized to focus on the most related HRL. For all four LRLs, the percentage of their most related HRL starts to decrease as training continues. For `aze`, DDS quickly comes back to using its most related HRL. However, for `bel`, DDS continues the trend of using all four languages. This shows that DDS is able to maximize the benefits of the multi-lingual data by having a more balanced usage of all languages.

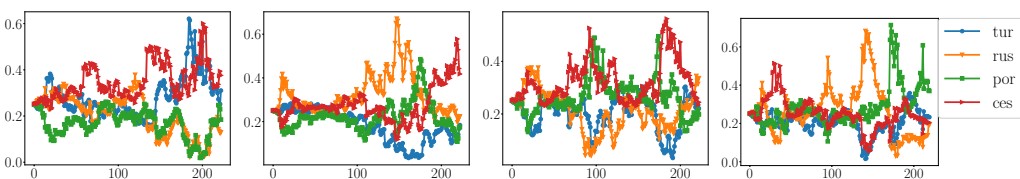

**Figure 5:** Language usage for DDS by training step. *From left to right*: `aze, bel, glg, slk`.

Figure 5 shows a more interesting trend of DDS without heuristic initialization. For both `aze` and `bel`, DDS focuses on the most related HRL after a certain number of training updates. Interestingly, for `bel`, DDS learns to focus on both `rus`, its most related HRL, and `ces`. Similarly for `slk`, DDS also learns to focus on `ces`, its most related HRL, and `rus`, although there is little vocabulary overlap between `slk` and `rus`. Also notably, the ratios change significantly over the course of training, indicating that different types of data may be more useful during different learning stages.

## 5 RELATED WORK

Many machine learning approaches consider how to best present data to models. First, difficulty-based curriculum learning estimates the presentation order based on heuristic understanding of the hardness of examples (Bengio et al., 2009; Spitkovsky et al., 2010; Tsvetkov et al., 2016; Zhang et al., 2016; Graves et al., 2017; Zhang et al., 2018; Platanios et al., 2019). These methods, though effective, often generalize poorly because they require task-specific difficulty measures. On the other hand, self-paced learning (Kumar et al., 2010; Lee & Grauman, 2011) defines the hardness of the data based on the loss from the model, but is still based on the assumption that the model should learn from easy examples. Our method does not make these assumptions. Closest to the learning to teach framework (Fan et al., 2018) but their formulation involves manual feature design and requires expensive multi-pass optimization. Instead, we formulate our reward using bi-level optimization, which has been successfully applied for a variety of other tasks (Colson et al., 2007; Anandalingam & Friesz, 1992; Liu et al., 2019a; Baydin et al., 2018; Ren et al., 2018).

Data selection for domain adaptation for disparate tasks has also been extensively studied (Moore & Lewis, 2010; Axelrod et al., 2011; Ngiam et al., 2018; Jiang & Zhai, 2007; Foster et al., 2010; Wang et al.). These methods generally design heuristics to measure domain similarity. Submodular optimization (Kirchhoff & Bilmes, 2014; Tschiatschek et al., 2014) selects training data that are similar to dev set, but the criterion is often based on hand-designed features and the data usage is predefined before training. Besides domain adaptation, selecting also benefits training in the face of noisy or otherwise undesirable data (Vyas et al., 2018; Pham et al., 2018).

Our method is also related to works on training instance weighting (Sivasankaran et al., 2017; Ren et al., 2018; Jiang & Zhai, 2007; Ngiam et al., 2018). These methods reweigh data based on a manually computed weight vector, instead of using a parameterized neural network. Notably, Ren et al. (2018) tackles noisy data filtering for image classification, by using meta-learning to calculate a locally optimized weight vector for each batch of data. In contrast, our work focuses on the general problem of optimizing data usage. We train a parameterized scorer network that optimizes over the entire data space, which can be essential in preventing overfitting mentioned in Sec. 2; empirically our method outperform Ren et al. (2018) by a large margin in Sec. 4. (Wu et al., 2018; Kumar et al., 2019; Fang et al., 2017) propose RL frameworks for specific natural language processing tasks, but their methods are less generalizable and requires more complicated featurization.

# 6 CONCLUSION

We present Differentiable Data Selection, an efficient RL framework for optimizing training data usage. We parameterize the scorer network as a differentiable function of the data, and provide an intuitive reward function for efficiently training the scorer network. We formulate two algorithms under the DDS framework for two realistic and very different tasks, image classification and multilingual NMT, which lead to consistent improvements over strong baselines.

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

# A  APPENDIX

## A.1  DERIVING GRADIENT OF $\psi$ FOR DIFFERENT OPTIMIZERS

First, we rewrite the update rule of $\theta$ in Eqn. 4 to incorporate the effect of its specific optimization algorithm.

For a fixed value of $\psi$, $J(\theta, \psi)$ can be optimized using a stochastic gradient update. Specifically, at time step $t$, we update

$$\theta_t \leftarrow \theta_{t-1} - g\big(\nabla_\theta J(\theta_{t-1}, \psi)\big) \tag{8}$$

where $g(\cdot)$ is any function that may be applied to the gradient $\nabla_\theta J(\theta_{t-1}, \psi)$. For instance, in standard gradient descent $g(\cdot)$ is simply a linear scaling of $\nabla_\theta J(\theta_{t-1}, \psi)$ by a learning rate $\eta_t$, while with the Adam optimizer (Kingma & Ba, 2015) $g$ also modifies the learning rate on a parameter-by-parameter basis.

Due to the relationship between $\theta_t$ and $\psi$ as in Eqn 8, $J(\theta_t, \mathcal{D}_{\text{dev}})$ is differentiable with respect to $\psi$. By the chain rule, we can compute the gradient $\nabla_\psi J(\theta_t, \mathcal{D}_{\text{dev}})$ as follows:

$$
\begin{aligned}
\nabla_\psi J(\theta_t, \mathcal{D}_{\text{dev}}) &= \nabla_{\theta_t} J(\theta_t, \mathcal{D}_{\text{dev}})^\top \cdot \nabla_\psi \theta_t(\psi) && \text{(chain rule)} \\
&= \nabla_{\theta_t} J(\theta_t, \mathcal{D}_{\text{dev}})^\top \cdot \nabla_\psi \big(\theta_{t-1} - g\big(\nabla_\theta J(\theta_{t-1})\big)\big) && \text{(substitute } \theta_t \text{ from Eqn 8)} \\
&\approx -\nabla_{\theta_t} J(\theta_t, \mathcal{D}_{\text{dev}})^\top \cdot \nabla_\psi g\big(\nabla_\theta J(\theta_{t-1})\big) && \text{(assume } \nabla_\psi \theta_{t-1} \approx 0)
\end{aligned}
\tag{9}
$$

Here, we make a Markov assumption that $\nabla_\psi \theta_{t-1} \approx 0$, assuming that at step $t$, given $\theta_{t-1}$ we do not care about how the values of $\psi$ from previous steps led to $\theta_{t-1}$. Eqn 9 leads to a rule to update $\psi$ using gradient descent:

$$\psi_{t+1} \leftarrow \psi_t + \eta_\psi \nabla_{\theta_t} J(\theta_t, \mathcal{D}_{\text{dev}})^\top \cdot \nabla_\psi g\big(\nabla_\theta J(\theta_{t-1}, \psi_t)\big), \tag{10}$$

Here we first derive $\nabla_\psi g$ for the general stochastic gradient descent (SGD) update, then provide examples for two other common optimization algorithms, namely Momentum (Nesterov, 1983) and Adam (Kingma & Ba, 2015).

**SGD Updates.**  The SGD update rule for $\theta$ is as follows

$$\theta_t \leftarrow \theta_{t-1} - \eta_t \nabla_\theta J(\theta_{t-1}, \psi) \tag{11}$$

where $\eta_t$ is the learning rate. Matching the updates in Eqn 11 with the generic framework in Eqn 8, we can see that $g$ in Eqn 8 has the form:

$$g\big(\nabla_\theta J(\theta_{t-1}, \psi)\big) = \eta_t \nabla_\theta J(\theta_{t-1}, \psi) \tag{12}$$

This reveals a linear dependency of $g$ on $\nabla_\theta J(\theta_{t-1,\psi})$, allowing the exact differentiation of $g$ with respect to $\psi$. From Eqn 10, we have

$$
\begin{aligned}
&\nabla J(\theta_t, \mathcal{D}_{\text{dev}})^\top \cdot \nabla_\psi g\big(\nabla_\theta J(\theta_{t-1}, \psi)\big) \\
&= \eta_t \cdot \nabla_\psi \mathbb{E}_{x,y \sim p(X,Y;\psi)} \left[ J(\theta_t, \mathcal{D}_{\text{dev}})^\top \cdot \nabla_\theta \ell(x, y; \theta_{t-1}) \right] \\
&= \eta_t \mathbb{E}_{x,y \sim p(X,Y;\psi)} \left[ \big( J(\theta_t, \mathcal{D}_{\text{dev}})^\top \cdot \nabla_\theta \ell(x, y; \theta_{t-1}) \big) \cdot \nabla_\psi \log p(x, y; \psi) \right]
\end{aligned}
\tag{13}
$$

Here, the last equation follows from the log-derivative trick in the REINFORCE algorithm (Williams, 1992).

**Momentum Updates.**  The momentum update rule for $\theta$ is as follows

$$
\begin{aligned}
m_t &\leftarrow \mu_t m_{t-1} + \eta_t \nabla_\theta J(\theta_{t-1}, \psi) \\
\theta_t &\leftarrow \theta_{t-1} - m_t,
\end{aligned}
\tag{14}
$$

where $\mu_t$ is the momentum coefficient and $\eta_t$ is the learning rate. This means that $g$ has the form:

$$
\begin{aligned}
g(x) &= \mu m_{t-1} + \eta_t x \\
g'(x) &= \eta_t
\end{aligned}
\tag{15}
$$

Therefore, the computation of the gradient $\nabla_\psi$ for the Momentum update is exactly the same with the standard SGD update rule in Eqn 13.

**Adam Updates.** We use a slightly modified update rule based on Adam (Kingma & Ba, 2015):

$$
\begin{aligned}
g_t &\leftarrow \nabla_\theta J(\theta_{t-1}, \psi) \\
v_t &\leftarrow \beta_2 v_{t-1} + (1 - \beta_2) g_t^2 \\
\hat{v}_t &\leftarrow v_t / (1 - \beta_2^t) \\
\theta_t &\leftarrow \theta_{t-1} - \eta_t \cdot g_t / \sqrt{\hat{v}_t + \epsilon}
\end{aligned}
\tag{16}
$$

where $\beta_2$ and $\eta_t$ are hyper-parameters. This means that $g$ is a component-wise operation of the form:

$$
\begin{aligned}
g(x) &= \frac{\eta_t \sqrt{1 - \beta_2^t} \cdot x}{\sqrt{\beta_2 v_{t-1} + (1 - \beta_2) x^2 + \epsilon}} \\
g'(x) &= \frac{\eta_t \sqrt{1 - \beta_2^t} (\beta_2 v_{t-1} + \epsilon)}{\left(\beta_2 v_{t-1} + (1 - \beta_2) x^2 + \epsilon\right)^{3/2}} \approx \eta_t \sqrt{\frac{1 - \beta_2^t}{\beta_2 v_{t-1}}},
\end{aligned}
\tag{17}
$$

the last equation holds because we assume $v_{t-1}$ is independent of $\psi$. Here the approximation makes sense because we empirically observe that the individual values of the gradient vector $\nabla_\theta J(\theta_{t-1}, \psi)$, *i.e.* $g_t$, are close to 0. Furthermore, for Adam, we usually use $\beta_2 = 0.999$. Thus, the value $(1 - \beta_2) x^2$ in the denominator of Eqn 17 is negligible. With this approximation, the computation of the gradient $\nabla_\psi$ is almost the same with that for SGD in Eqn 13, with one extra component-wise scaling by the term in Eqn 17.

### A.2 Hyperparameters for multilingual NMT

In this section, we give a detailed description of the hyperparameters used for the multilingual NMT experiments.

- We use a 1 layer LSTM with hidden size of 512 for both the encoder and decoder, and set the word embedding to size 128.
- For multilingual NMT, we only use the scorer to model the distribution over languages. Therefore, we use a simple 2-layer perceptron network as the scorer architecture. Suppose the training data is from $n$ different languages. For each target sentence and its corresponding source sentences, the input feature is a $n$-dimensional vector of 0 and 1, where 1 indicates a source language exists for the given target sentence.
- We simply use the dev set that comes with the dataset as $\mathcal{D}_{\text{dev}}$ to update the scorer.
- The dropout rate is set to 0.3.
- For the NMT model, we use Adam optimizer with learning rate of 0.001. For the distribution parameter $\psi$, we use Adam optimizer with learning rate of 0.0001.
- We train all models for 20 epochs without any learning rate decay.
- We optimize both the NMT and DDS models with Adam, using learning rates of 0.001 and 0.0001 for $\theta$ and $\psi$ respectively.

### A.3 Dataset statistics for Multilingual NMT

| LRL | Train | Dev | Test | HRL | Train |
|-----|-------|-----|------|-----|-------|
| aze | 5.94k | 671 | 903 | tur | 182k |
| bel | 4.51k | 248 | 664 | rus | 208k |
| glg | 10.0k | 682 | 1007 | por | 185k |
| slk | 61.5k | 2271 | 2445 | ces | 103k |

**Table 2:** Statistics of the multilingual NMT datasets.

### A.4 Hyperparameters for image classification

In this section, we provide some additional details for the image classification task:

- We use the cosine learning rate decay schedule (Loshchilov & Hutter, 2017), starting at $0.1$ for CIFAR-10 and $3.2$ for ImageNet, both with $2000$ warmup steps.

- For image classification, we use an identical network architecture with the main model, but with independent weights and a regressor to predict the score instead of a classifier to predict image classes.

- To construct the $\mathcal{D}_{\text{dev}}$ to update the scorer, we hold out about 10% of the *training* data. For example, in CIFAR-10 (4,000), $\mathcal{D}_{\text{dev}}$ is the last 400 images, while in ImageNet-10%, since we use the first 102 TFRecord shards, $\mathcal{D}_{\text{dev}}$ consists of the last 10 shards. Here, "last" follows the order in which the data is posted on their website for CIFAR-10, and the order in which the TFRecord shards are processed for ImageNet. All data in $\mathcal{D}_{\text{dev}}$ are excluded from $\mathcal{D}_{\text{train}}$. Thus, for example, with CIFAR-10 (4,000), $|\mathcal{D}_{\text{train}}| = 3600$, ensuring that in total, we are only using the amount of data that we claim to use.

- We maintain a moving average of all model parameters with the rate of $0.999$. Following Kornblith et al. (2019), we treat the moving statistics of batch normalization (Ioffe & Szegedy, 2015) as *untrained parameters* and also add them to the moving averages.

- For ImageNet, we use the post-activation ResNet-50 (He et al., 2016). The batch sizes for CIFAR-10 and for ImageNet are $128$ and $4096$, running for 200K steps and 40K steps, respectively.

