# OpenReview forum: "Optimizing Data Usage via Differentiable Rewards"
_ICLR.cc/2020/Conference — Reject_

### Official Review · AnonReviewer1 · 2019-10-17
**Official Blind Review #1**

**Rating:** 3

**Review:**

The paper proposes an iterative method that jointly trains the model and a scorer network that places a non-uniform distribution over data sets.  The paper proposes a gradient method to learn the scorer network based on reinforcement learning, which is novel as to what the reviewer knows.

There are several concerns/questions:

1) The paper doesn’t define the D_{dev} clearly. How is D_{dev} chosen? Is it a subset of D_{train}?

2) In section 2.1, why “smaller development set D_{dev} is much closer to the P_{test}(X,Y)”? P_{test}(X,Y) is supposed to be not observed during training?

3) In Eq (5), if D_{dev} is s subset of D_{train}, if \theta* is the minimal of J, it means the gradient
at  \theta* is 0. To calculate the gradient of J with respect to \psi, by chain rule, it need to calculate gradient to \theta* first then \theta* to \psi. If gradient of \theta* is 0, the product is also 0? So the \psi will not be updated if D_{dev}  is sufficiently similar to D_{train} ?

4) In Section 2.3, it omits the second order Hessian term. How does that influence the performance?

5) it mentions “without significant computing overhead“ in abstract, which is not demonstrated elsewhere.

6) In the experiments, table 1, it seems the major improvement comes from retrain and TCS rather than DDS? In figure 3, it is better to show the weights of an image without DDS and comparing that with DDS.

7) The paper contains many typos such as Eqn.11 is not defined in the main paper, the “Eqn ??” Appears in the appendix, “tha minimizes” etc.

In general, the idea of the paper is natural and the results seem promising. I am looking forward to the reply to my questions/concerns.

#############

I have read the author's feedback. I think the clarity of both methodology and experiment does not reach the acceptance level and would maintain my current rating.


**Experience Assessment:**

I have read many papers in this area.

**Review Assessment: Checking Correctness Of Derivations And Theory:**

I assessed the sensibility of the derivations and theory.

**Review Assessment: Checking Correctness Of Experiments:**

I assessed the sensibility of the experiments.

**Review Assessment: Thoroughness In Paper Reading:**

I read the paper at least twice and used my best judgement in assessing the paper.

---

> ### Author Response · Authors · 2019-11-07
> **Response to reviewer #1**
>
> Thank you very much for providing many pieces of good feedback and clarification questions. We believe that in both the response and the revised draft we have clarified or rectified all of the reservations that were stated in the original review. We would appreciate if you could check our response and revisions (and if these have indeed clarified the concerns, revise the overall assessment). We would also be happy to continue the discussion and make any additional modifications as deemed necessary.
>
> Reviewer #1 question 1): The paper doesn’t define the D_{dev} clearly. How is D_{dev} chosen? Is it a subset of D_{train}?
>
> Response:
> For our machine translation tasks, $D_{dev}$ is simply the dev set that comes with the dataset.
>
> For our image classification tasks, for $D_{dev}$ we hold out about 10% of the *training* data. For example, in CIFAR-10 (4,000), $D_{dev}$ is the last 400 images, while in ImageNet-10%, since we use the first 102 TFRecord shards, $D_{dev}$ consists of the last 10 shards. Here, “last” follows the order in which the data is posted on their website for CIFAR-10, and the order in which the TFRecord shards are processed for ImageNet. All data in $D_{dev}$ are excluded from $D_{train}$. Thus, for example, with CIFAR-10 (4,000), $|D_{train}| = 3600$, ensuring that in total, we are only using the amount of data that we claim to use.
>
> Thus, in all cases, there is no overlap between $D_{train}$ and $D_{dev}$, or $D_{test}$ and $D_{dev}$ (as is standard in machine learning experiments). We have added a clarification in the method section of the paper, and we also updated the details in the appendix.
>
> Reviewer #1 question 2): In section 2.1, why “smaller development set D_{dev} is much closer to the P_{test}(X,Y)”? P_{test}(X,Y) is supposed to be not observed during training?
>
> Response:
> It is correct that P_{test}(X, Y) is not observed, but practically in model training it is commonly more possible to collect a dev set that reflects the test scenario. To take the example of the multilingual NMT, in this case we would like to use training data from *many different languages* to improve the performance of *a particular low-resource language*. Here, D_{train} is the aggregation of data from all languages, while D_{dev} could be a separate small set of data from the low-resource language we are interested in. This small dev set is possible to gather, even if we can’t gather a large training set in the language. P_{test}(X, Y) in this case is the distribution of the low-resource language, which is much better captured by the small D_{dev} from this low-resource language.
>
> Similar settings can easily be thought of in other scenarios as well: in a domain adaptation setting we can obtain a small dev set in the target domain, or in a setting of training on noisy data we can often obtain a small clean dev set. We have updated the method section of the paper to clarify this issue.
>
> Finally, even if the training set and dev set come from *exactly* the same data distribution, likelihood on the dev set is still going to be a better estimator of test performance, as the model is not able to train on the dev set directly (which is why we use dev sets in standard machine learning setups in the first place).
>
> Reviewer #1 question 3): In Eq (5), if D_{dev} is s subset of D_{train}, if \theta* is the minimal of J, it means the gradient at  \theta* is 0. To calculate the gradient of J with respect to \psi, by chain rule, it need to calculate gradient to \theta* first then \theta* to \psi. If gradient of \theta* is 0, the product is also 0? So the \psi will not be updated if D_{dev}  is sufficiently similar to D_{train} ?
>
> Response:
> As we mentioned in point (1), $D_{dev}$ does not overlap with $D_{train}$, so these gradients will be inherently different.

---

> > ### Author Response · Authors · 2019-11-07
> > **Continue response to Reviewer #1**
> >
> > Reviewer #1 question 4): In Section 2.3, it omits the second order Hessian term. How does that influence the performance?
> >
> > Response:
> > Our gradient derivation in Eqn. 6 uses the Markov assumption that in the previous step $\psi_{t-1}$ is already updated with regard to $\theta_{t-1}$, so the effect of $\psi$ on $\theta_{t-1}$ is likely to be minimal. This assumption can simplify and speed up computation. Moreover, this allows us to have a natural interpretation of the update rule for the data scorer: it should up-weight the training data that have similar gradient direction with the dev data.
> >
> > The use of the Markov assumption is based on its use and empirical success in previous work on bi-level optimization, such as Hyper Gradient Descent [1] and many others. Of course, this is a simplifying assumption, but we believe that our empirical results show that the proposed method is useful nonetheless.
> >
> > Relaxing this assumption would be an interesting avenue for future work. However at the same time how to do so without resulting in large increases in complexity, both with respect to difficulty in implementation,and with respect to computation/memory complexity, is a challenge that would require additional methodological advantages beyond the scope of the current work.
> >
> >
> > Reviewer #1 question 5): it mentions “without significant computing overhead“ in abstract, which is not demonstrated elsewhere.
> >
> > Response:
> > In Section 4.2, we describe the nominal increase in training time for DDS. Please see our full response to Reviewer #3, Question 2.
> >
> > Reviewer #1 question 6) In the experiments, table 1, it seems the major improvement comes from retrain and TCS rather than DDS? In figure 3, it is better to show the weights of image without DDS and comparing that with DDS.
> >
> > Response:
> > In table 1, the retrained-DDS is still using DDS. We train a scorer with the model until convergence using DDS, then reinitialize the model and train both the scorer and the model again using DDS. Essentially, the first pass of the DDS training moves the scorer parameters to have a good prior distribution over the training data, so that the second DDS training pass is able to improve even further.
> > For the multilingual training, TCS+DDS simply initializes the scorer distribution with the TCS distribution before training using DDS. We compare TCS+DDS with the best baseline, including using only TCS. This shows that DDS brings significant gains over TCS for all four languages.
> > The description of retrained-DDS and TCS+DDS can be found at the end of section 4.1 In the paper.
> >
> > Reviewer #1 question 7): The paper contains many typos such as Eqn.11 is not defined in main paper, the “Eqn ??” Appears in appendix, “tha minimizes” etc.
> >
> > Response:
> > Thank you very much for pointing out the typos and providing other good feedback. We have corrected the typos and updated the paper along with the appendix.
> >
> > [1] Online learning rate adaptation with hypergradient descent  https://arxiv.org/abs/1703.04782

---

### Official Review · AnonReviewer2 · 2019-10-20
**Official Blind Review #2**

**Rating:** 6

**Review:**

This paper presents a reinforcement learning approach towards using data that present best correlation with a validation set’s gradient signal. The broader point of this paper is that there is inevitably some distribution shift going from train to test set - and the validation set can be a small curated set whose distribution is closer to the testing distribution than what the training dataset's distribution is.

The problem setup bears relationship to several areas including domain adaptation/covariate shift problems, curriculum learning based approaches amongst others. One assumption that I see which needs to be understood more is equation (6) - wherein, somehow, there is a Markov assumption used to zero out the contribution of the scoring network on parameters unto previous time step. Trying to understand the implications of this assumption (how the performance varies with/without this assumption) would be instructive for understanding potential shortcomings of this framework.

I think the paper is well written, handles an important question. That said, I am not too aware of recent work in this area to make a decisive judgement on this paper’s novelty/contributions.

**Experience Assessment:**

I do not know much about this area.

**Review Assessment: Checking Correctness Of Derivations And Theory:**

N/A

**Review Assessment: Checking Correctness Of Experiments:**

I assessed the sensibility of the experiments.

**Review Assessment: Thoroughness In Paper Reading:**

I read the paper at least twice and used my best judgement in assessing the paper.

---

> ### Author Response · Authors · 2019-11-07
> **Response to reviewer #2**
>
> We thank the reviewer for providing the feedback and suggestions. Please see our response to Reviewer #1, question 4). We have also updated the paper to add some clarifications. We would really appreciate if you could check whether our response has cleared your concern, and that you could consider improving the overall assessment. We would love to continue the discussion and make improvements to our paper.

---

> > ### Comment · AnonReviewer2 · 2019-11-14
> > **Response to the authors**
> >
> > Thank you for your response - I read through your note to Reviewer 1. I stick to my current assessment and would defer to other reviewers/AC towards the decision. Thanks!

---

### Official Review · AnonReviewer3 · 2019-10-26
**Official Blind Review #3**

**Rating:** 6

**Review:**

Summary: This paper introduces a simple idea to optimize the weights of a weighted empirical training distributions. The goal is to optimize the population risk, and the idea is to optimize a distribution over the training examples to maximize the cosine similarity between training set gradients and validation set gradients. The distribution over the training set is parameterized by a neural network taking as arguments the

Strengths:
- The method is quite simple.
- The results appear to be strong, although I am less familiar with the NMT baselines. The imagenet results seem quite strong to me.

Weaknesses:
- I couldn't find a particularly clear description of the scoring networks architecture. Given that it observes the whole dataset, this seems like a critical choice that could have a big impact on the complexity of this approach. At the very least, this should be clearly reported, and I recommend a more thorough investigation of this choice.
- The authors report that their method takes 1.5x to 2x longer to run than the uniform baseline. Yet, they ran all methods for the same number of steps / epochs. It seems to me that a fairer comparison might be letting all methods enjoy the same total budget measure roughly by wall time.

Questions:
- I didn't follow why the computation of the per example gradient grad l(x_i, y_i, theta_t-1) is so onerous. Isn't that computed on line 5 already?

**Experience Assessment:**

I have read many papers in this area.

**Review Assessment: Checking Correctness Of Derivations And Theory:**

I assessed the sensibility of the derivations and theory.

**Review Assessment: Checking Correctness Of Experiments:**

I assessed the sensibility of the experiments.

**Review Assessment: Thoroughness In Paper Reading:**

I read the paper at least twice and used my best judgement in assessing the paper.

---

> ### Author Response · Authors · 2019-11-07
> **response to reviewer #3**
>
> We thank the reviewer for providing many good suggestions and questions. We have addressed your concerns here and updated the paper with some clarifications. We would appreciate if you could check our response and revisions (and if these have indeed clarified the concerns, revise the overall assessment). We would also be happy to continue the discussion and make any additional modifications as deemed necessary.
>
> reviewer #3 question1: scoring network architecture
>
> response:
> We are sorry for the lack of clarity with respect to this! This was simply an oversight.
>
> For image classification, we use an identical network architecture with the main model, but with independent weights and a regressor to predict the score instead of a classifier to predict image classes. For the multilingual NMT experiments, since we only want to model a simple distribution over n training languages, we use a fully connected 2-layer perceptron network. For each target sentence and its corresponding source sentences, the input feature is a n-dimensional vector of 0 and 1, where 1 indicates a source language exists for the given target sentence. We have updated the image classification and NMT instantiation section, as well as the appendix, with clarifications of the network structure, and will release our code once the paper is accepted.
>
>
> reviewer #3 question2: The authors report that their method takes 1.5x to 2x longer to run than the uniform baseline. Yet, they ran all methods for the same number of steps / epochs. It seems to me that a fairer comparison might be letting all methods enjoy the same total budget measure roughly by wall time.
>
> response:
> The main objective of DDS is to improve model performance, while remaining much simpler and more efficient than other methods that optimize a data selector using reinforcement learning that require multiple independent training runs. For example, in the IMDB movie review experiment in  [1], the data filtering agent is also trained for 200 episode, where each episode uses around 40% of the whole dataset, requiring a total of 80x more training time than a single training run. Therefore, the 1.5-2x increase in time afforded by DDS is much more manageable.
>
> For image classification, training the standard baseline for longer does not help, since the main model will start to overfit, which indicates that spending more time on the baseline would not have a positive effect.
>
> reviewer #3 question3:  I didn't follow why the computation of the per example gradient grad l(x_i, y_i, theta_t-1) is so onerous. Isn't that computed on line 5 already?
>
> response:
> In practice, a single gradient is computed with respect to a mini-batch of training data of size n to improve computational efficiency. However, using the per-example gradient requires one to compute the gradient for each example in a batch, which essentially slows down training by a factor of n. Therefore, we propose the simplification in Eqn. 7 to compute the per example gradient.
>
>
> [1] Learning what data to learn https://arxiv.org/pdf/1702.08635.pdf

---

### Decision · Program_Chairs · 2019-12-19

**Decision:**

Reject

**Comment:**

The paper proposes an iterative learning method that jointly trains both a model and a scorer network that places a non-uniform weights on data points, which estimates the importance of each data point for training.  This leads to significant improvement on several benchmarks.  The reviewers mostly agreed that the approach is novel and that the benchmark results were impressive, especially on Imagenet.  There were both clarity issues about methodology and experiments, as well as concerns about several technical issues.  The reviewers felt that the rebuttal resolved the majority of minor technical issues, but did not sufficiently clarify the more significant methodological concerns. Thus, I recommend rejection at this time.